# Bayesian-Maximum-Entropy Reweighting of IDP Ensembles Based on NMR Chemical Shifts

**DOI:** 10.3390/e21090898

**Published:** 2019-09-17

**Authors:** Ramon Crehuet, Pedro J. Buigues, Xavier Salvatella, Kresten Lindorff-Larsen

**Affiliations:** 1Institute for Advanced Chemistry of Catalonia (IQAC-CSIC) c/Jordi Girona 18-26, 08034 Barcelona, Spain; pedrojuanbj@gmail.com; 2Structural Biology and NMR Laboratory, Linderstrøm-Lang Centre for Protein Science, Department of Biology, University of Copenhagen, 2200 Copenhagen, Denmark; 3Institute for Research in Biomedicine (IRB Barcelona), Barcelona Institute of Science and Technology, Baldiri Reixac 10, 08028 Barcelona, Spain; xavier.salvatella@irbbarcelona.org; 4ICREA, Passeig Lluís Companys 23, 08010 Barcelona, Spain

**Keywords:** Bayesian methods, maximum entropy, chemical shifts, intrinsically disordered proteins, protein ensembles, structural modelling, NMR, molecular dynamics

## Abstract

Bayesian and Maximum Entropy approaches allow for a statistically sound and systematic fitting of experimental and computational data. Unfortunately, assessing the relative confidence in these two types of data remains difficult as several steps add unknown error. Here we propose the use of a validation-set method to determine the balance, and thus the amount of fitting. We apply the method to synthetic NMR chemical shift data of an intrinsically disordered protein. We show that the method gives consistent results even when other methods to assess the amount of fitting cannot be applied. Finally, we also describe how the errors in the chemical shift predictor can lead to an incorrect fitting and how using secondary chemical shifts could alleviate this problem.

## 1. Introduction

Intrinsically Disordered Proteins (IDPs) do not fold into stable conformations, because their free energy landscapes possess many shallow local minima. Some of these may correspond to conformations rich in secondary structure and α-helices appear to be the most common of these regular structures. Their ability to fold into different conformations allows IDPs to interact with different binding partners and tuning their populations by post-translational modifications can allow for the regulation of important cellular functions. An accurate description of the secondary structures adopted by IDPs, as well as of their populations, can therefore help us understand the cellular behaviour of this important class of proteins.

Several methods exist to generate ensembles of configurations representing the conformational heterogeneity of IDPs, including all-atom molecular dynamics simulations [1], implicit-solvent Monte Carlo simulations [2] and other sampling approaches [3,4,5]. As the time evolution of the conformations is not relevant in the types of ensemble reweighting we here consider, from now on we loosely refer to molecular dynamics as any simulation method that can produce an ensemble of conformations. In practice, the simulated ensembles differ from the true ensembles for three main reasons. First, because the simulation conditions may not be exactly the same as the experimental ones—for example, different ionic strength, salts or pH may be used. Second, because equilibrium sampling of the IDPs conformational space is challenging to achieve, mainly due to its enormous size. And third, because all energy functions used in molecular simulations describe particle interactions with limited accuracies. This is particularly so for water-protein interactions, that are prevalent in IDPs due to their large solvent-accessible surface area [1,6,7,8,9,10]. Also, because of their “flat” energy landscapes, IDPs may be particularly prone to even minor inaccuracies in energy functions. Thus, it would be desirable to correct the simulated ensembles by improving their agreement to measured experimental data.

Some experimental data produce properties that depend on the molecular structure, and therefore can be used to correct simulated ensembles [11,12,13]. However, the resolution of the experimental data is lower than that of molecular dynamics so that determining conformational ensembles from experiments needs to deal with experimental uncertainties [14,15]. There are basically two different approaches to compare experiments and simulations. Either molecular coordinates are produced from the experimental data, as is done with X-ray crystallography or NMR models [16], and these can then be compared against the simulated structures; or the experimental observable is generated from the simulated conformations and compared directly to the experimentally measured quantities [17]. The first approach inevitably introduces some models and assumptions to generate a higher resolution model (molecular coordinates) from a lower resolution one (experimental outcome). Therefore, the second approach is less model dependent as long as the observable quantities can be calculated from the molecular coordinates. The latter is achieved with a predictor also known as a forward model. The need for fast and (relatively) accurate predictors is a limitation of these second approach but they exist for several nuclear magnetic resonance (NMR) techniques.

NMR chemical shifts (CS) are extremely useful to give insight into the secondary structure of IDPs [18,19,20,21,22,23]. CS measured by NMR are determined mainly by the type of residue but, for a given residue, some atoms will give different CSs depending on the local geometry of the residue, as quantified for example by its location in the Ramachandran map [23]. CS for a given protein conformation can also be predicted with a low computational cost. Therefore, CS can be used to benchmark conformational ensembles produced by molecular dynamics simulations and, if necessary, to fit them to experiment.

Maximum Entropy or Bayesian approaches are powerful tools to fit simulated ensembles to experimental CSs. Maximum Entropy produces the minimum perturbation to the original ensemble so that it fits the experimental data. In a pure Maximum Entropy approach, experimental and prediction errors are not taken into account. This limitation can be overcome by reformulating the Maximum Entropy approach within a Bayesian reasoning [24,25,26]. In this work, we will use the algorithm called Bayesian/Maximum Entropy (BME) put forward by Bottaro et al. [26], which is based on work by Hummer and Köfinger [25]. The CS calculated from a simulated conformational ensemble can be considered the prior distribution in a Bayesian formulation. Even if the experimental error is known, errors in the prior are difficult to quantify, including force field accuracy, lack of full convergence of the simulations and errors in the forward model used to calculate CS from the ensemble. In the BME reasoning, we can tune the confidence in the prior with respect to the likelihood of the data with a subjective hyper-parameter θ (see Equation (1) in Methods).

The choice of the confidence parameter θ is far from trivial. θ defines how much we rely on the simulated results compared to the experimental data; i.e., how much we want to reweight the simulated data to fit the experimental ones [12,27]. Here we suggest a new technique to determine this number when fitting simulated ensembles of an IDP using CS.

Protein CS have previously been used in Bayesian approaches to structure determination [28,29,30,31,32,33]. Purely Bayesian methods require parameter sampling and are computationally expensive. Therefore, they cannot easily be applied directly to ensembles of thousands of structures. The ensembles are usually clustered into a much smaller set, but, as clustering of IDPs is far from trivial, here we use BME which can deal with tens of thousands of structures and their experimental observables. The calculations reported in this work took at most a few minutes to run - depending on the number of θ values scanned - in a Python Jupyter notebook running on a workstation.

As an example of an IDP we will use the protein ACTR [34]. ACTR consists of 71 residues, and was recently used as a benchmark system in a series of molecular dynamics simulations with different force fields [35]. We here focus on the simulations with three force fields: AMBER99SB-disp, AMBER03ws, and CHARMM36m (respectively named a99SBdisp, a03ws and C36m hereafter). As previously described [35], different force fields predict different helical content for different regions of the protein sequence (Figure 1) These force fields represent some of the newest versions of two popular force field families [1]. However, our aim here is not to evaluate their strengths or weaknesses, but to use them as three possible conformational ensembles.

Methods to test approaches to combine experiments and simulations are often hampered by the fact that we generally do not know the true target ensemble, and thus it is then not clear what an accurate result would be. Thus, to gain full control of the reweighting procedure and to know the desired outcome, we here use synthetic data. This is an approach that has proved powerful for discovering the impact of experimental uncertainties in structure determination that we and others have previously used [36,37,38,39,40,41]. In particular, we consider the a99SBdisp as a reference ensemble for generating synthetic data (i.e., we treat calculated CS from this ensemble as the experimental data). To clarify that these structures and their predicted CS are not actually experimental, we call this ensemble *target ensemble*. Then we will fit the predicted CS of other ensembles to the CS of the target ensemble. Namely, we will fit the a03ws and the C36m trajectories, which correspond to different conformational ensembles. We will also fit the a99SBdisp derived CS into the target ensemble, i.e., into itself, to evaluate the effect of the errors of the CS predictor (see the Methods section).

The use of CS derived from a target ensemble which is known will allow us to compare the reweighted distribution of CS to the target-ensemble distribution, which would not be accessible from an actual NMR experiment. The CS from an NMR experiment depend on the time-scale of exchanging conformations. If, as it is for IDPs, this time-scale is fast, the measured CS is a time and ensemble average of all conformations. Thus we only get the first moment (the mean) of the distribution of CS in the conformational ensemble. In cases when the experiment reports distributions or higher distribution moments, several approaches can extend the Maximum Entropy method to those cases [42,43,44].

Even if the experiment only produces the mean of the distribution, the goal of reweighting is to obtain an ensemble that better reproduces the experimental ensemble of structures, not only its (measured) mean. By using synthetic data, we can compare how the distribution of CS during the reweighting approaches the target distribution. After all, the BME procedure guarantees that we can systematically approach the mean value, but at some point, overfitting could lead to a distribution that deviates from the target ensemble distribution instead of approaching it.

In this work we will explore how similar the target and reweighted ensembles are during a reweighting procedure using the distance between the distributions (see Methods section). We will also compare the behaviour of the average value of these distributions. From that, we will propose some rules to assess the amount of reweighting (determining θ) in real situations where the target distribution is unknown because only the experimental average value is available.

## 2. Methods

The MD trajectories with different force fields were generated as part of a recent study aimed at developing and benchmarking force fields [35] and were kindly shared by the authors. In this work we use the trajectories generated with the Amber ff03ws force field used with the TIP4P/2005 water model and with scaled protein-water interactions (a03ws) [6], the CHARMM36m (C36m) force field with its native TIP3P water model [45], and Amber ff99SB-disp with its native TIP4P-D-based water model (a99SBdisp) [35]. All the simulations are 30 µs long and contain *N* = 29777 frames. In this work we take the a99SBdisp as the target distribution and we will fit the a03ws and the C36m to this one, using CS. In the main text we focus on the a03ws results, but all the analysis is reproduced for the C36m ensemble as supplementary figures.

CS can be used to reweight IDP ensembles because they are sensitive to local secondary structure conformations. Indeed, experimentally derived CS indices allow for the quantification of secondary structure based on the CS values [46,47]. Although one might use the predicted secondary structures from these algorithms to reweight simulated ensembles this would introduce unnecessary model bias, and we instead computed the CSs from the ensembles and calculate the primary data directly. For that, we used existing algorithms and software to predict CS from structures.

Methods to calculate CS can introduce both systematic and random errors in their predicted values, but because we do not have experimental CS for our target ensemble, but predicted ones, we need a procedure to model these errors, that would be present in an actual experiment. The use of a predictor with perfect precision and accuracy can give some insight into the information content of CS, as we also explore below, but it does not reflect an experimental setting. Although we could have arbitrarily introduced random and systematic error to the predicted CS of the target ensemble, we preferred an alternative approach where the errors came from using two different predictors. We believe that that magnitude of the introduced errors will be more realistic than chosen arbitrarily.

We used Sparta+ [48] and PPM [49] to simulate the target a99SBdisp CS and only PPM [49] to simulate the reweighted distributions. When not stated otherwise, the target CS come from Sparta+ and we therefore do not expect that reweighting will lead to perfect agreement with the CSs predicted from the ensembles using PPM. PPM was specifically designed to calculate CS for individual frames in a conformational ensemble, i.e., it does not include in its training the thermal fluctuations around a given structure that others methods implicitly absorb into the predicted values. We have used as the error for the PPM predictor the reported errors of the validation set (Table 3 in [49]), namely 1.06 for Cα, 1.23 for Cβ and 1.32 for the carbonyl C. This choice is based on the fact that the experimental error in the CS is negligible compared to the error of the forward model, and thus that the main uncertainty when determining whether a predicted average CS is “close” to the experimental measurement comes from the error of the forward model. We note also that the estimated error of the CS predictors come from analyses of a much more heterogeneous set of structures and might be an overestimate compared to the relatively narrow distribution of CS in IDPs. Our set of CS contained 69 Cα chemical shifts, 64 Cβ chemical shifts and 69 C chemical shifts, totalling *m* = 202 chemical shift data.

The application of the BME approach is equivalent to the minimization of the following objective function:(1)$$ℒ=\frac{m}{2}\chi_{red}^{2}\left(w_{1}\cdots w_{N}\right)-\theta S_{REL}\left(w_{1}\cdots w_{N}\right)$$
where *w_i_* are the reweighted weights, *N* is the number of structures in the ensemble and *m* the number of experimental observables. $\chi_{red}^{2}$ measures the degree of fitting:(2)$$\chi_{red}^{2}=\frac{1}{m}\overset{m}{\underset{i=1}{\sum }}\frac{{\left(\sum_{i=1}^{N}w_{j}CS_{j}^{i}-CS_{EXP}^{i}\right)}^{2}}{\sigma_{i}^{2}}$$
where $CS_{j}^{i}$ is the chemical shift of atom *i* in structure *j* and $CS_{EXP}^{i}$ is the target value. And the relative entropy, defined as $S_{REL}=-\sum_{j=1}^{N}w_{j}\log\left(\frac{w_{j}}{w_{j}^{0}}\right)$ measures the deviation from the initial weights, $w_{j}^{0}$, in our case taken as 1/N. *S_REL_* is closely related to the effective sample size, *N_eff_*, a useful measure of how much reweighting has taken place (see Equation (3)).

In the calculation of $\chi_{red}^{2}$ we divide by the number of CS data, assuming completely independent data. Each $\sigma_{i}^{}$ is equated to the error of the PPM predictor reported above, as errors in the convergence of the simulations and the accuracy of the force fields are difficult to quantify. Because of these assumptions, the absolute value of $\chi_{red}^{2}$ is of limited utility and it should be merely interpreted as a normalized measure of the residuals.

The effective sample size varies from 1 when all structures have the same weight, i.e., before the reweighting; to 0, in case a single structure gets 100% of the weight. There are different definitions on the effective sample size, one commonly used was put forward by Kish [50], but here we use the definition based on relative entropy, as it fits better within the BME paradigm:(3)$$N_{eff}=\exp\left(S_{REL}\right)$$

The optimization of Equation (1) can be performed with a set of auxiliary Lagrange parameters λ, which allow writing the optimal weights as [24,51,52]:(4)$$w_{i}=\frac{1}{Z}\exp\left(-\overset{m}{\underset{j=1}{\sum }}\lambda_{j}CS_{j}^{i}\right)$$
where *Z* is a normalization constant. This expression highlights two important aspects of the BME approach. First, the *N* weights are determined by optimizing the values of *m* λ parameters. As *N*, the number of structures in the ensemble, is usually much larger than the number of experimental data *m*, this reduces overfitting. Second, the fitting consists of adding a linear term to the free energy associated to each observable. This linear term can be seen as the minimal perturbation needed to tilt the distribution of the computed ensemble so that its average shifts towards the experimental value. Indeed, a purely Bayesian approach introducing a linear term to the energy [53] leads to results that are equivalent to a Maximum Entropy approach including an error term [54]. The strength of the reweighting is directly determined by the values of λ, and not by θ. In other words, a given θ may lead to different λ, and thus different reweightings for two different ensembles fitted to the same target values. (see Appendix A)

After each of the fittings, we compared the average values with Equation (2) and the distribution of values between the reweighted and the target distribution with the Wasserstein distance, as implemented in the Python package SciPy [55]. As each pair of reweighted and target CS distributions produces a Wasserstein distance value, thus we used the mean of all these *m* values as a measure of the similarity between two sets of distributions.

Secondary CSs were calculated for the target and reweighted ensemble in the same way. The secondary structure of each conformation was determined with the DSSP [56] implementation in MDTraj [57]. Then, the averaged CS for residues in coiled conformations represents the random coil reference. The secondary CS is the result of subtracting the random coil reference to the CS.

## 3. Results

The first step in a reweighting procedure is to ensure that the ensemble can be reweighted. Ideally, the ensemble should contain values around the target value. Appendix A shows that this is the case for the two fitted ensembles discussed in this work. If the target value lies outside the distribution of the calculated ensemble values, the convergence of the reweighting is compromised. In that case one should check if there are systematic errors in the predictor, the experiment, or the simulation. Alternatively increasing the sampling could populate the less probable regions of the distribution. Sampling of these regions can also be achieved by running simulations with restraints [33,52,58,59,60,61,62].

The improvement of the fit resulting from the reweighting can be measured with the $\chi_{red}^{2}$ explained in the Methods section. Lower values of $\chi_{red}^{2}$ indicate a better fitting. As reweighting takes place, some conformations gain weight while others lose it. This leads to a reduced entropy compared to the original ensemble, which can be converted to an effective sample size, *N_eff_*, ranging from 1 to 0. As the amount of reweighting increases to fit the data more accurately, *N_eff_*, decreases. In some systems, the shape of the $\chi_{red}^{2}$ vs. *N_eff_* curve allows for the determination of a critical value from which no further reweighting is necessary [26,63]. However, in fitting CS of ACTR we find a relatively homogeneous decrease of $\chi_{red}^{2}$, which makes it difficult to decide when to stop fitting(see Figure 2). A second common procedure is the Wald-Wolfowitz run test [64] for the residuals, where one checks that they are mutually independent. Long stretches of residues with the same sign indicate that more reweighting is possible. However, when using BME, decreasing θ decreases the size of the residuals, but not their signs (see Appendix A), i.e., each reweighted CS approaches its target CS from the same side as θ is decreased. This renders the Wald-Wolfowitz run test procedure inadequate.

We note here that Figure 2 shows a low $\chi_{red}^{2}$ even at the start of the fitting procedure. A $\chi_{red}^{2}$ below one may suggest that the fit is good enough, but can also reflect problems in estimating errors in both experiments and forward models, and in the statistical independence of the data. The $\sigma_{i}^{}$ used in the calculation corresponds to the PPM error for a heterogeneous set of structures and their experimental CS. When compared to Sparta+ ACTR chemical shifts, the errors are much smaller (See Appendix A). Because of this error underestimation, the absolute value of $\chi_{red}^{2}$ is of limited use. If we use the PPM CS for the target ensemble, we are effectively using a predictor without error. In that case, $\chi_{red}^{2}$ is ill-defined, but we get a curve very similar to Figure 2 (see Appendix A). This suggests that the CS difference reflects more the different structural composition of the ensemble (Figure 1) than the errors of the predictor.

Indeed, the lack of a clear target value for $\chi_{red}^{2}$ is one key reason why we need to determine the parameter θ. Independently of the origin of the small $\chi_{red}^{2}$ the ensembles described by the three force fields are different (Figure 1) and should be distinguishable by their differences in CS. For many residues the CS differences are small because they present a random coil structure, but the region where the target and simulated ensembles show different helical content, the CS show larger differences (Figure 3 and Appendix A). This trend is clearer for the a03ws force field than for the C36m (Appendix A). This suggests that reweighting would improve the simulated ensembles by rendering them closer to the target ensemble. But traditional methods do not always provide a clear answer to how to choose θ to avoid over-fitting but have enough reweighting.

In this work we suggest using a validation set to determine the relative weights of the experimental data and the simulation prior when reweighting, i.e., determine an optimal value of θ. A similar idea was used by Cesari et al. to find the optimal balance between the distribution arising from the force field and the experimental data [65]. Here, we put forward the following procedure:(1)Split the frames of the trajectory into a training set (*t*) and a validation set (*v*) of the same size. We used odd and even frames for the training and validation set respectively. We use sets of the same size because we aim to compare average values and distributions, not individual conformations of the validation set. A validation set as large as the test set is therefore needed to minimize the standard error of the mean and discretization errors of the distribution. We here chose to use an interleaved training and validation set after confirming that the two sets have highly uncorrelated CSs as frames are sampled only once every 1 ns.(2)Fit the BME for a range of θ values to the training set and apply the optimized Lagrange parameters λ to reweight the validation set.(3)For each of the θ values, evaluate the following properties
$\chi_{red}^{2}$ (Equation (2)) of the training ($\chi_{red}^{2}\left(t\right)$) and the validation set ($\chi_{red}^{2}\left(v\right)$).Average distance between the training and the validation sets (*D*(*t*,*v*)).Average distance between the training set and the target (goal) distribution (*D*(*t*,*g*)), and average distance between the validation set and the target (goal) distribution (*D*(*v*,*g*)).

Note that step 3c is only possible in the case of synthetic data and is used as a benchmark for a selection procedure. Ideally we would like to reweight as long as the validation set distribution approaches the target distribution, i.e., we want to minimize *D*(*v,g*). In a real case scenario, as the target distribution is not known, we cannot measure this distance. Therefore we need to see if any of the accessible measures can give us information about *D*(*v,g*).

Figure 4 plots all the values described in the previous procedure. As expected, the BME produces an average CS that approaches the target CS, so that it systematically reduces $\chi_{red}^{2}\left(t\right)$. At θ ≈ 3 the validation set fits best the target CS and then $\chi_{red}^{2}\left(v\right)$ starts increasing. This is a sign of overfitting. Interestingly the behaviour of these averaged values is followed very similarly by the distributions, as represented by the shape of *D*(*t,g*) and *D*(*v,g*). The fact that the distributions, which are what we aim to fit, and the average values have parallel behaviours is very positive as this tells us that we can use the average value as a proxy of the distribution. At a value close to θ ≈ 3, *D*(*t,v*) has a high curvature and starts growing. The training and the validation distributions, which had been very similar for values θ > 3, start diverging. This confirms that at this point, overfitting occurs. The fit to the C36m ensemble shows a very similar trend (Appendix A). A different measure also suggests the start of the overfitting regime. Appendix A shows the root mean square of the Lagrange λ terms that the BME introduces for the fit (see Equation (4)). At θ < 2 the λ parameters start growing at a much faster speed, showing that from this point the reweighting becomes very strong and very sensitive to the θ value.

To evaluate the importance of the error introduced by the predictor and whether this was leading to overfitting, we reproduced the same analysis with target CS coming from the PPM predictor, i.e., excluding the predictor error. As Appendix A depicts, the behaviour of the different quantities is equivalent to Figure 4, with the only difference than the difference in distributions *D(t,g)* and *D(v,g)* and shifted to lower values, as expected from the use of the same predictor. The two distributions are more similar, because the predictor error is missing, and they become diverging at a slightly smaller value of θ, as $\chi_{red}^{2}$ also does. This shows that the stochastic error of the predictor averages out and that BME is robust to it. Systematic errors in the predictor could be important, as will be shown below, but in this case their influence is much smaller than the differences in CS arising from conformational differences.

Once an optimal value for θ has been determined, other properties of the ensemble can be calculated. As an example, here we calculate the α-helical content and plot it in Figure 5 and Appendix A. We stress that the reweighting improves the fit, but does not lead to a perfect agreement. There are several sources for the disagreement. The major source of the disagreement is the fact that the predictor (PPM software) has an error in the prediction with respect to the target value (Sparta+ software). Even for the same ensemble of configurations predicted and target CS differ, thus leading to a disagreement between secondary structure content, as is discussed below. Further, although the two are tightly related, there is not a one-to-one relationship between the secondary structure and the CS values. Also, the reweighted ensemble may not contain structures with the optimal α-helical content. For example, the a03ws ensemble contains a negligible amount of α-helical structures around residue number 60, therefore, even after the reweighting, the helicity of this region cannot increase and remains too low. Finally, as long as the experimental data does not uniquely specify the target distribution, procedures such as this will always be affected by the choice of a reference (prior) distribution and will thus not give the exact target distribution.

When we examine the α-helical content of the validation ensemble after reweighting we find that it is further away from the initial ensemble than the training ensemble. This makes sense as we are using a Maximum Entropy algorithm, which ensures minimal perturbation of the training ensemble from the original one. The minimal perturbation, expressed as minimization of the Kullback-Leibler divergence, is true only for the training ensemble. Therefore, the parameters of the training ensembles applied to the validation ensemble lead to a further divergence from the original ensemble. Consequently, the described procedure should be used to determine the optimal θ, but once it is known, one should re-fit the complete ensemble with that optimal θ.

Figure 5 and Appendix A show that the reweighting changed the helicity the most in the region between residues 30 and 45. This is because CSs are not only sensitive to local residue conformations, but also to neighbouring residue conformations. This results in the calculated data in long helical stretches differing most from a random coil than short helices. Consequently, the reweighting is able better to ‘see’ long helices and reweight those regions accordingly.

The reweighting will only improve properties that are sensitive to the experimental data used. In this case, the CS of different atoms are sensitive to the helical conformations to a different degree. Thus, reweighting leads to an ensemble with improved helical content. Other properties, such as the radius of gyration (*R_g_*), are sensitive to the global mass distribution of the conformations. For expanded ensembles, an increase of helicity leads to more collapsed ensembles, as helical structures are compact [66,67]. For the a03ws, reweighting leads to an overall decrease of helicity (Appendix A), and therefore to a larger *R_g_*. For the C36m, the slight increase in helicity does not correlate with the *R_g_*, presumably because the initial ensemble is already rather compact. As also emphasized by Best as co-workers, both local (such as secondary structure) and global (such as *R_g_*) properties should be used to characterize IDPs ensembles, and one should not expect the reweighting based on one set of these properties to improve the other [1]. Using these other properties as cross-validation, as it is sometimes done, may lead to an incorrect perception of the amount of reweighting needed. As an example, the C36m fitting shows a minimum that could suggest that the evolution of *R_g_* could be used as a cross-validating property, but the a36m shows a monotonic increase, for the reasons previously mentioned. If several experimental data are known, they should be included in the reweighting to obtain more realistic ensembles with improved local and global properties.

A good way to estimate the effect of the predictor error is to fit the target ensemble to itself. In that case, the disagreement between predicted and target CS comes exclusively from the use of different predictors. We will refer to predictor error as the difference between the predictor used for the reweighting ensemble (PPM) and the predictor used as a target (Sparta+) without any assumption of which one gives CS closer to the true experimental ones.

If the predictor error was a Gaussian noise with zero mean, it would average to zero for an ensemble of tens of thousands of structures. However, the predictor has systematic deviations that depend on the residue type and its conformation (Appendix A). The systematic deviations also correspond to what higher helical content would give: negative C and CA and positive CB (see Figure 3). This suggests that the reweighting will tend to decrease the helical content.

By following the same procedure as before, we obtain an optimal value of θ ≈ 2 (Appendix A). It may seem surprising that when the reweighted ensemble matches exactly the target ensemble the θ value is lower than when fitting the a03ws and C36m ensembles. Two things need to be considered. First, the amount of disagreement between the CSs is not much lower than when fitting the a03ws or the C36m ensembles. This shows that the predictor is a major source of errors, but even with considerable unknown systematic errors in the predictor, the reweighting of the a03ws and C36m ensembles gave consistently improved ensembles. Second, θ values between ensembles are not comparable because θ does not determine the strength of the reweighting. The strength of the reweighting is determined from the optimized values of the weights, which themselves arise from the optimized values of the Lagrange (λ) parameters (see Equation (4)). For a given θ, the reweighting is weaker for the current a99SBdisp ensemble than for the a03ws and C36m (see Appendix A). As Figure 6 shows, even after the reweighting with θ = 2, the helicity has changed, by at most ~ 0.05, from 0.31 to 0.26 in the region of residue 37. For all the other regions the changes in helicity are smaller. After all, the reweighting procedure ‘sees’ CSs that would correspond to lower helicity, and therefore results in an ensemble with lower helicity.

Although the reweighting from the target a99SBdisp ensemble was not large, deviating from the initially correct ensemble is not satisfactory. We therefore looked for alternatives to minimize the reweighting. One alternative is to improve the matching between calculated and target CS. One would be to use the best possible predictor as its accuracy when comparing with actual experimental CS should not be overlooked.

Considering that a source of error in the predictor may come from the baseline CS for each residue, we hypothesized that the use of secondary CSs could lead to a cancellation of errors. Secondary CS are defined as the difference between measured CSs and CSs from random coil conformations [20]. They measure the change of CS when a given residue adopts a secondary structure conformation. If predictors are good at describing this change then secondary CS could lead to more sensitive reweightings. As secondary structures can be assigned at a residue level for each conformation of a computational ensemble, the generation of secondary CS from computed ensembles is fast and simple (see Methods section).

The use of secondary CS leads to essentially no-reweighting. We repeated the optimization procedure described above with calculated and target secondary CSs. The optimization of θ leads to flat curves (Appendix A), that at first sight could suggest that the optimal value of θ is difficult to determine. To a certain extent this is true, the reason being that the reweighting is mostly insensitive to θ. What is more interesting is that the reweighting is negligible even for extremely small values of θ, as Appendix A shows, so that the reweighted ensemble remains essentially unchanged and thus, close to target ensemble as it was desirable.

As expected, the negligible reweighting does not lead to overfitting. Appendix A shows that in the fitting of secondary CS, the effective sample size stabilizes to a value of 0.42 for θ < 10^−3^. A value of *N_eff_* far from 0 shows that the method is not overfitting. We remind the reader that the amount of reweighting is not determined by θ itself but by the Lagrange λ parameters that result from the optimization procedure. The result of this procedure tells us that, whatever our relative confidence in the computed ensemble and the target CS, the reweighting will be essentially zero.

The secondary CS before any reweighting already show an excellent agreement with the target ones, so that reweighting is not necessary. The $\chi_{red}^{2}$ before reweighting in one order of magnitude smaller for secondary CS than for CS. However, we wanted to test our method in an extreme case. We conclude that even when it could easily lead to overfitting, the nature of BME prevents that, leading to stable results. But it also shows that when there are alternative methods to reduce the errors of the predictors, as the use of secondary CSs, the reweighting becomes simpler (in the sense that one does not need to tune θ) and more accurate.

## 4. Discussion

The Maximum Entropy approach is a simple, yet powerful method to fit computed ensembles to experimental data [68]. Its extension into a Bayesian approach allows the treatment of uncertainties arising from both experiment and computation [17,25,26]. It is often difficult to determine the optimal balance between the prior information, encoded in the force field, and the experimental data. These difficulties arise in particular because computational errors are difficult to assess. They arise from the predictor (the forward model), the convergence of the simulation, the accuracy of the force field used and the computational reproducibility of the experimental conditions. Also, in many cases it is difficult to determine accurate estimates of experimental errors. Together, these effects result in difficulties in determining a unique value of θ, or that a wide range of possible θ values appear comparably good. Because of uncertainties in the experimental and computational errors it is difficult to choose the level of reweighting simply by the magnitude of $\chi_{red}^{2}$, thus making it important to find objective and robust criteria to determine the values of θ.

Here we suggest using a validation set that arises from the same distribution, and therefore belongs to the same model as the training set. As we show in the case of comparing results from CS and radius of gyration, the use of other physical observables for cross-validation can sometimes lead to a wrong choice of θ. If the observables are strongly correlated to the ones being fitted, it will be hard to detect any overfitting, as the overfitted ensemble, even if having a small effective size, will also reproduce the correlated observables. On the contrary, if the physical observable is not correlated, then there is no reason why the reweighted ensemble should improve its fitting, as we showed for the radius of gyration.

The split of the data between training and validation sets can be done in different ways. The two sets would be maximally uncorrelated if the sets arise from the first and the second half of the trajectory. But unless each of the halves is fully equilibrated—which is not the case in our trajectory—the two sets would correspond to different distributions, and one should not expect that the same Lagrange λ parameters should apply to the two sets. For instance, if the first half samples more helical conformations than the second, the reweighting could lead to a reduction of helicity for the first half and an increase for the second. On the contrary, by using interleaved frames to create training and validation sets we ensure they represent samples of the same distribution, whether fully equilibrated or not. Therefore, the same Lagrange λ parameters should improve both training and test until the moment we start overfitting the training set. Therefore, there is a range of θ values before the overfitting that are dependent on the distribution to be fitted but independent of the particular structures chosen from that distribution. This is the basis of our selection of the optimal θ value. However, if the validation set is strongly correlated to the training set, the overfitting could not be detected in the validation set. We therefore recommend checking the correlation between neighbouring frames. Alternatively, the reweighting and validation suggested procedure can be repeated for a subsample of the trajectory and it should lead to a similar optimal θ value.

Even if we do not incur in overfitting, a large amount of reweighting indicates either that the prior is considerably different from the target distribution or that the predictor has important systematic errors. In both cases the reweighted ensemble will still be far from the actual ensemble, and one should be especially cautious in not over-interpreting it, especially in extracting structural information from it that is not sensitive to chemical shifts, as we have discussed for the radius of gyration.

We also showed that the errors of the predictor should not be underestimated. In the specific case of NMR CS, it appears that even when the predictors were designed and parameterized to remove systematic errors, they may still arise in applications to a particular system. These unknown systematic errors contribute to the reweighting and are a major source of overfitting that our method avoids to a certain extent.

The use of secondary CS seems a promising approach to reduce the errors of the predictors. Its calculation from a computational ensemble is simple, but the experimental calculation is not trivial. One approach is to use tabulated CS for protein random coils [20], including corrections depending on neighbouring residues as well as pH and temperature [69,70,71,72,73]. A more precise but more time consuming approach is to completely denature an IDP and measure the resulting CS [74,75]. Both approaches introduce some uncertainties that may render the secondary CS less accurate. In future work we will explore whether they still remain a better alternative when reweighing computational ensembles of IDPs.

We would like to end this discussion by reminding the reader that the BME and related approaches are particularly suited to avoid overfitting. Maximization of the entropy implies that the reweighted distribution is minimally perturbed, unlike in other reweighting approaches [45,46,47,48,49,50,51,52,53]. One may think that still, small values of θ, which means a strong confidence on the experimental data, would lead to an overfit of the finite number of computed structures. However, the fitting procedure fits a small number of parameters (equal to the number of CSs), in our case *m* = 202, to a much larger number of structures, *N* = 29977. If each weight was fitted individually, the procedure would be highly prone to overfitting, with *N* = 29777 parameters. Instead, BME determines weights from a much smaller set of parameters, the number of observables (*m* = 202) plus θ. The effect of this is that the possibility of overfitting is substantially reduced.

## 5. Conclusions

In this work we have shown how chemical shifts can be used to improve the configurations arising from molecular dynamics simulations of intrinsically disordered proteins. We have introduced a systematic method to assess the amount of reweighting (θ) needed to fit the experimental chemical shifts based on cross-validation. This approach allows to circumvent the difficulty of knowing the errors associated to the simulations and the chemical shift predictors. We have also shown how the predictor errors lead to an incorrect reweighting as they include a systematic bias. This error seems to be greatly reduced by using secondary chemical shifts, something that we will further explore in future work.

## Figures and Tables

**Figure 1 entropy-21-00898-f001:**
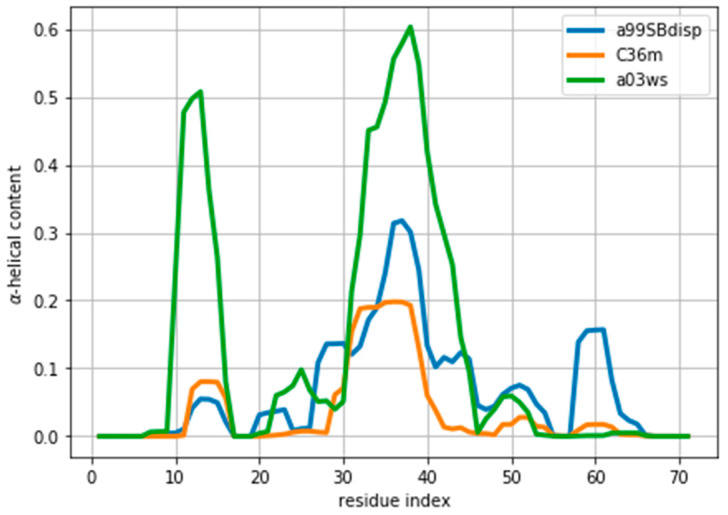
Amount of helical content in the ACTR trajectories simulated using different force fields. The a99SBdisp is taken as the target ensemble and the C36m and a03ws as the simulated trajectories to reweight.

**Figure 2 entropy-21-00898-f002:**
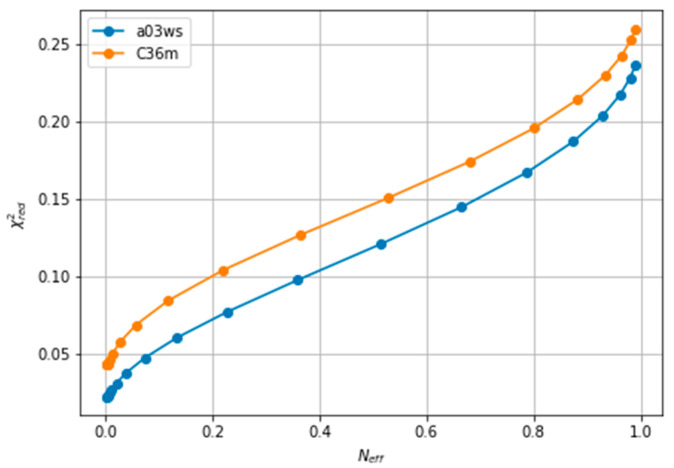
Evolution of $\chi_{red}^{2}$ with the effective sample size (*N_eff_*), showing the lack of an L-shaped curve. The a99SBdisp is taken as the target ensemble and the C36m and a03ws as the simulated trajectories to reweight.

**Figure 3 entropy-21-00898-f003:**
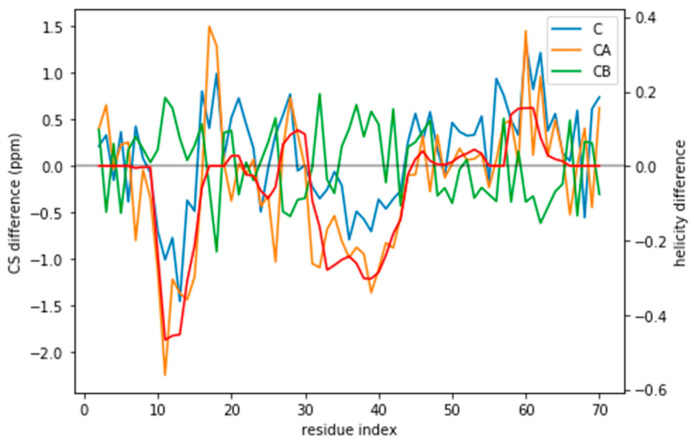
CS difference between the target CS and the a03ws CS for the three atoms used (C, CA, and CB) compared to the difference in helicity for these two ensembles (in red). Although for many residues the difference lies below the error of the predictor, for some regions it does not. Besides, the helicity is correlated with the CA and the CB CS difference.

**Figure 4 entropy-21-00898-f004:**
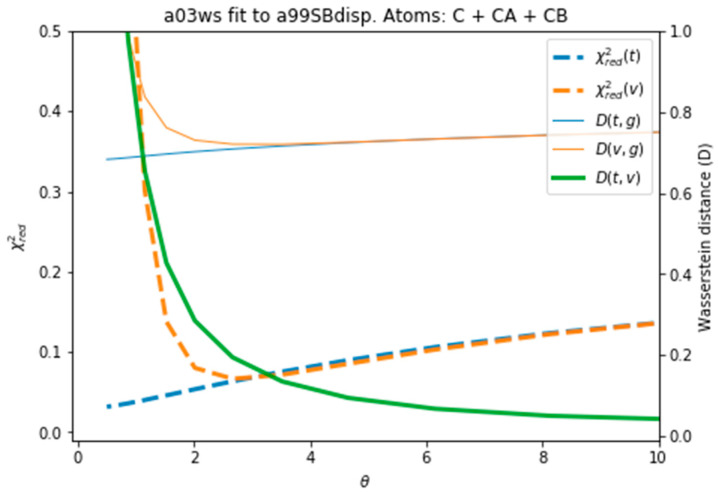
Behaviour of different quantities during the reweighting procedure of the a03ws ensemble. The quantities defined as thin lines would not be measurable in a real case scenario, but their behaviour can be inferred from the quantities in thick lines.

**Figure 5 entropy-21-00898-f005:**
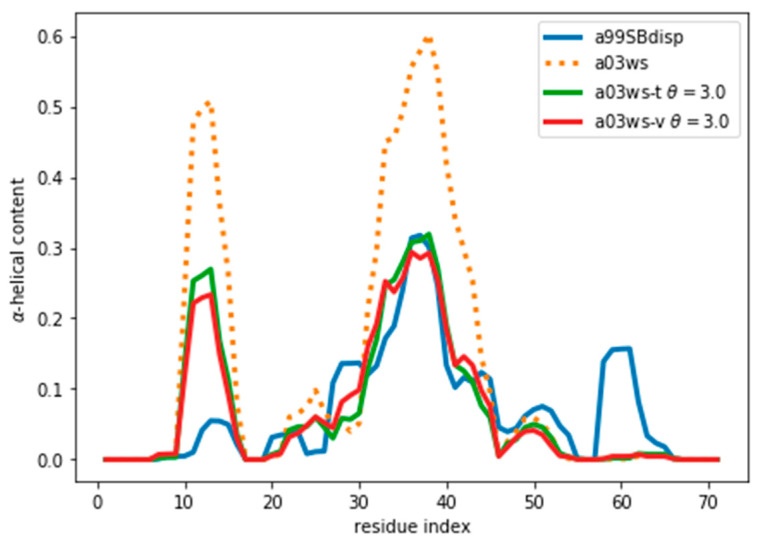
α-Helical content for the reweighted and target ensembles. For the reweighted ensembles the train (a03ws-t) and validation (a03ws-v) sets are shown. The original ensemble before the reweighting is also shown.

**Figure 6 entropy-21-00898-f006:**
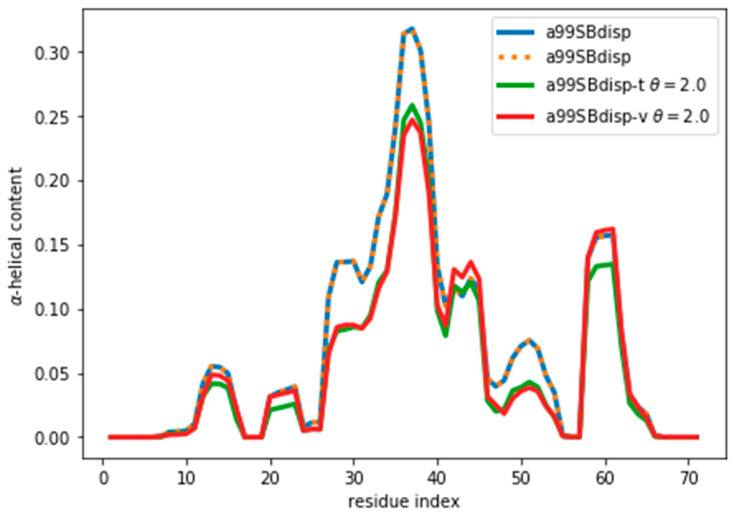
α-Helical content for the reweighted and target ensembles. For the reweighted ensembles the train (a99SBdisp-t) and validation (a99SBdisp-v) sets are shown. The original ensemble before the reweighting is also shown and, as expected, it corresponds exactly to the target ensemble as they are the same.

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
