# Peer review of "Bayesian-Maximum-Entropy Reweighting of IDP Ensembles Based on NMR Chemical Shifts"

_entropy, 2019, doi:10.3390/e21090898_

Round 1

Reviewer 1 Report

The paper by Crehuet et al. describes the outcome of maximum entropy reweighting of IDP trajectories based on simulated chemical shift data from one trajectory which is held as a reference. The study is very well designed and very well described. I have some comments about some aspects of the calculations which may not trivial for the average reader and that, in my opinion, should be discussed. I here mark them as major revisions, because additional calculations might be required.

1a) A rather important point is with the use of the reduced chi-squared. I do not want to question the validity of the chi-squared for chemical shifts, which do not have a gaussian distribution of uncertainty (this, however, could be mentioned). However, it is very important to stress that a reduced chi-squared lower than 1 implies overfitting (cpr Taylor, J. Introduction to Error Analysis, the Study of Uncertainties in Physical Measurements, 2nd Edition). This may also explain the lack of "L-shape" in figure 2. 

1b) The work of Hummer's group suggest that theta values larger than two imply that too much reweighting is needed, therefore invalidating the choice of the prior distribution - i.e.: the prior overestimates the content of the distribution in some parts of the conformational space whereas others are not sampled (cpr figure 9 in the reference at point 2a). This is already discussed by the authors, but should be made clearer. This is likely reflected in the accumulation of weights into fewer conformations and could be easily checked by a histogram of the weights, which could be provided for each reweighting.

1c) points 1a and 1b likely imply that the error in the forward model is dominating the reconstruction. I would therefore recommend (see point 2) to combine the use of different predictors, to bring the "experimental" average values closer to the "center" of the data space (again cpr. figure 9 in the reference at point 2a), which would be fairer to the expected behavior of maximum entropy reweightings.

2) chemical shift prediction is afflicted by systematic errors. One, rather common way, to overcome this limitation is to use different predictors (e.g.: shiftx, camshift...). In the present design, it should be possible to use different predictors at least in two different ways:

a) use the spread of the calculated values as uncertainty.

b) for each residue, use the best fitting predictor in the reweighting (this, of course, comes at the price of increasing the computational time of the reweighting).

3) In other papers using chemical shifts, H-alphas, amide protons and amide nitrogens are also used. I may agree that the use of amide protons for IDPs would be prone to larger errors, therefore these should probably be left out, but the others are good sources of information would be easily accessible experimentally, and therefore could lend a helping hand to the reconstruction.

Minor points:

1) I have no particular feeling for the wording "amount of fitting" that, at least in the abstract, I would call "quality of fitting"

2) overall, literature referencing could be more comprehensive. I here give some examples that are particularly strinking to me, but I invite the authors to check by themselves if some other relevant work could be included: 

a) page 2, line 50-59: this paragraph gives a nice introduction to the concepts of ensemble reconstruction. It would be nice if the authors could cite the other recent review on this topic, which appears not to be particularly well known, as opposed to references 13-14:  Ravera et al., Physical Chemistry Chemical Physics 18 (8), 5686-5701

b) Use of chemical shifts for ensemble reconstruction in IDPs is discussed extensively in Fisher et al. J. AM. CHEM. SOC. 2010, 132, 14919. This paper must be cited. 

Author Response

The paper by Crehuet et al. describes the outcome of maximum entropy reweighting of IDP trajectories based on simulated chemical shift data from one trajectory which is held as a reference. The study is very well designed and very well described. I have some comments about some aspects of the calculations which may not trivial for the average reader and that, in my opinion, should be discussed. I here mark them as major revisions, because additional calculations might be required.

We thank the reviewer for her/his careful reading of the paper and his valuable comments. Hereafter we answer his comments and concerns one by one. We believe that the corrected version has improved in solidity and coherence thanks to the reviewer’s comments.

1a) A rather important point is with the use of the reduced chi-squared. I do not want to question the validity of the chi-squared for chemical shifts, which do not have a gaussian distribution of uncertainty (this, however, could be mentioned). However, it is very important to stress that a reduced chi-squared lower than 1 implies overfitting (cpr Taylor, J. Introduction to Error Analysis, the Study of Uncertainties in Physical Measurements, 2nd Edition). This may also explain the lack of "L-shape" in figure 2. 

The question of overfitting is very relevant for a procedure such as the one we suggest. However, we believe that the apparent overfitting based on the value of the reduced chi-square is probably due to an overestimation of the predictor error. Indeed, although we use the reported error for PPM, the CS difference between values predicted from Sparta+ and PPM is considerable smaller than 1ppm. This is probably because the error was calculated on a set of structures much more heterogeneous than our. Indeed, the problem with accurate estimations of errors from both forward model and experiments is one of the reasons the hyperparameter, Theta, is necessary. 

We have included a figure in the SI to show these results and included these points in the text.

1b) The work of Hummer's group suggest that theta values larger than two imply that too much reweighting is needed, therefore invalidating the choice of the prior distribution - i.e.: the prior overestimates the content of the distribution in some parts of the conformational space whereas others are not sampled (cpr figure 9 in the reference at point 2a). This is already discussed by the authors, but should be made clearer. This is likely reflected in the accumulation of weights into fewer conformations and could be easily checked by a histogram of the weights, which could be provided for each reweighting.

We have extended our discussion on the choice of theta and the possibility of overfitting. It is a bit unclear what the reviewer means by theta>2. In general, the greater the value of theta, the less reweighting. We, however, agree that if a very small value of theta is needed, this may well indicate a poor prior that needs extensive reweighting. But we disagree that there is a particular value of theta (such as 2) that determines “too much” reweighting (in part because we don’t always have good measures of the errors of the forward model or experiments; see below), and in any case one does not always have the luxury of having a prior that needs only little reweighting. Also, as we show in the case of secondary chemical shifts, the lagrange parameters, and not theta, is what determines the amount of reweighting. Further, because if we have enough coverage of all the regions of the prior, important reweighitngs can be performed without incurring in overfitting. However it is true that the reweighted ensemble should not be over-interpreted, in particular if extensive rewrighting is applied to a poor prior, and we have clarified that in the revised discussion. 

1c) points 1a and 1b likely imply that the error in the forward model is dominating the reconstruction. I would therefore recommend (see point 2) to combine the use of different predictors, to bring the "experimental" average values closer to the "center" of the data space (again cpr. figure 9 in the reference at point 2a), which would be fairer to the expected behavior of maximum entropy reweightings.

We have explored the influence of the error of the forward model by repeating a fit with a zero-error predictor. We have also improved the text describing the error of the predictor (when compared to the reference predictor). 

The new results show that the systematic error of the predictor cannot be overlooked, but that the reconstruction when using different ensemble is dominated by their different secondary structure content (see Figures S4 and S8).

2) chemical shift prediction is afflicted by systematic errors. One, rather common way, to overcome this limitation is to use different predictors (e.g.: shiftx, camshift...). In the present design, it should be possible to use different predictors at least in two different ways:

a) use the spread of the calculated values as uncertainty. b) for each residue, use the best fitting predictor in the reweighting (this, of course, comes at the price of increasing the computational time of the reweighting).

We have calculated and plotted the spread of the calculated values (see figure S3) to explain that the forward model uncertainty is lower than reported. It is true that in a real-case scenario we could use different predictors and chose the best fitting, but in case of synthetic data this  would favour some predictors based on their similarity to the reference one, not on the merits of predicting experimental chemical shifts. We think that the comparison of different predictors falls beyond the scope of the present work. Further, while consensus prediction have their place, it is not trivial to find a suitable combination of truly independent predictors, and thus the variance across the predictors will depend substantially on this choice.

3) In other papers using chemical shifts, H-alphas, amide protons and amide nitrogens are also used. I may agree that the use of amide protons for IDPs would be prone to larger errors, therefore these should probably be left out, but the others are good sources of information would be easily accessible experimentally, and therefore could lend a helping hand to the reconstruction.

It is true that other chemical shifts would provide additional structural information, and therefore improve the quality of the fitted ensembles. However the aim of the paper was to suggest a way to selected theta in a Bayesian/Maximum-Entropy approach, and for that purpose we believe the choice of the data used is sufficient. 

There is also the practical limitation of the PPM software not producing predictions for H-alpha atoms.

Minor points:

1) I have no particular feeling for the wording "amount of fitting" that, at least in the abstract, I would call "quality of fitting"

Because we mean is how strong we fit or how much we fit, we have thus replaced "amount of fitting" with “extent of fitting”, which may represent better our idea than "quality of fitting".

2) overall, literature referencing could be more comprehensive. I here give some examples that are particularly strinking to me, but I invite the authors to check by themselves if some other relevant work could be included: 

a) page 2, line 50-59: this paragraph gives a nice introduction to the concepts of ensemble reconstruction. It would be nice if the authors could cite the other recent review on this topic, which appears not to be particularly well known, as opposed to references 13-14:  Ravera et al., Physical Chemistry Chemical Physics 18 (8), 5686-5701 b) Use of chemical shifts for ensemble reconstruction in IDPs is discussed extensively in Fisher et al. J. AM. CHEM. SOC. 2010, 132, 14919. This paper must be cited.

These and several other references have been added. We apologize for important omissions but this is an active field of research and are not attempting to review it comprehensively.

Reviewer 2 Report

Dr. Crehuet and collaborators present in their article "Bayesian-Maximum-Entropy reweighting of IDPs ensembles based on NMR chemical shifts" a new method assign NMR chemical shift to intrinsically disordered proteins. The results are clearly presented, the figures are present in good quality and with very good captions. The conclusion is supported by the results. I suggest the publication of this manuscript in the present form.

Author Response

Dr. Crehuet and collaborators present in their article "Bayesian-Maximum-Entropy reweighting of IDPs ensembles based on NMR chemical shifts" a new method assign NMR chemical shift to intrinsically disordered proteins. The results are clearly presented, the figures are present in good quality and with very good captions. The conclusion is supported by the results. I suggest the publication of this manuscript in the present form. 

We thank the reviewer for her/his positive evaluation of the manuscript. We have made several changes to the manuscript as suggested by the other reviewers and we encourage this reviewer to check the improved version of our work.

Reviewer 3 Report

In this work authors use the Bayesian Maximum Entropy approach to fit with synthetic NMR chemical shift data the molecular dynamics ensembles of an intrinsically disordered protein. The project design and the analysis of the results are rigorously presented in the manuscript and I found very useful the possibility of use this method for fitting computational ensembles directly from experimental raw data. Nevertheless, I consider that there are still minor issues in the work that can be improved in favor of a greater clarity.

1. I think the Introduction is the section that can be most improved. In particular, I found the paragraph of the justification of the use of NMR data in 50-66 lines (page 2) rather confusing. For instance, sentences such:

“The use of experimental data to correct the simulated ensembles adds another source of uncertainty [11,12]. Molecular dynamics simulations produce ensembles of conformations, often at a level of atomic detail, but the experimental data only provides an indirect and noisy representation of these conformations [13].”

can be misunderstood and give the wrong idea that the experimental data are more inaccurate than computational results (!!!), when the idea is much simpler, i.e. the resolution of the experimental data and the type of measured properties make them sometimes inadequate for fitting purposes. Moreover, the sentence: “Even for techniques with atomic resolution such as X-ray crystallography, the experimental outcome is a diffraction pattern, not a protein structure.”

can be removed, according to my opinion, as it is confusing and unnecessary to justify the use of NMR data.

2. The other issue that I found is concerning the choice of using two different CS predictors. Authors wrote in lines 136-138 (page 4):

“To account for the error of the predictor and generate results that mimic an experimental situation, we have predicted the CS of the target distribution and the reweighted distributions with different predictors.”

but as authors explained later in lines 323-325 the comparison between predictors can only account for the difference between both predictor results, and not for the true error of the predictor with respect to the experimental data. Besides, it is unclear for me what is the meaning of “mimicking an experimental situation” by using two different predictors. If the purpose of the research design is  “to gain full control of the reweighting procedure and to know the desired outcome” (lines 97-98), I think that the choice of employing two different CS predictors is contrary to this aim. Therefore, I strongly recommend the authors to clarify this point.

3. I could find some typos along the text:

lines 32-35:

“The ability to fold into different conformations allows IDPs to interact with different binding partners, and tuning their population these conformationsby post-translational modifications allows the regulation of their biological functions.”

line 284:

“...therefore, even after the reweighting, the helicity of this region cannot not increase and remains too...”

Captions of Figures S9 and S10 and in lines 475 and 477 of the manuscript: “...thinklines…”

Author Response

In this work authors use the Bayesian Maximum Entropy approach to fit with synthetic NMR chemical shift data the molecular dynamics ensembles of an intrinsically disordered protein. The project design and the analysis of the results are rigorously presented in the manuscript and I found very useful the possibility of use this method for fitting computational ensembles directly from experimental raw data. Nevertheless, I consider that there are still minor issues in the work that can be improved in favor of a greater clarity.

We thank the reviewer for his/her thorough reading of the paper and his/her positive evaluation. We have addressed his/her points as follows.

We have also made other changes to the manuscript as suggested by the other reviews and we encourage this reviewer to check the improved version of our work.

I think the Introduction is the section that can be most improved. In particular, I found the paragraph of the justification of the use of NMR data in 50-66 lines (page 2) rather confusing. For instance, sentences such:

“The use of experimental data to correct the simulated ensembles adds another source of uncertainty [11,12]. Molecular dynamics simulations produce ensembles of conformations, often at a level of atomic detail, but the experimental data only provides an indirect and noisy representation of these conformations [13].”

can be misunderstood and give the wrong idea that the experimental data are more inaccurate than computational results (!!!), when the idea is much simpler, i.e. the resolution of the experimental data and the type of measured properties make them sometimes inadequate for fitting purposes. 

We thank the reviewer and agree that this sentence could indeed be misunderstood as suggested. Several parts of the introduction have been re-written and in particular this paragraph, hopefully expressing the ideas more clearly.

Moreover, the sentence:

 “Even for techniques with atomic resolution such as X-ray crystallography, the experimental outcome is a diffraction pattern, not a protein structure.” can be removed, according to my opinion, as it is confusing and unnecessary to justify the use of NMR data.

We agree that this sentence was unnecessary and we have removed it.

The other issue that I found is concerning the choice of using two different CS predictors. Authors wrote in lines 136-138 (page 4):

“To account for the error of the predictor and generate results that mimic an experimental situation, we have predicted the CS of the target distribution and the reweighted distributions with different predictors.”

but as authors explained later in lines 323-325 the comparison between predictors can only account for the difference between both predictor results, and not for the true error of the predictor with respect to the experimental data. Besides, it is unclear for me what is the meaning of “mimicking an experimental situation” by using two different predictors. If the purpose of the research design is  “to gain full control of the reweighting procedure and to know the desired outcome” (lines 97-98), I think that the choice of employing two different CS predictors is contrary to this aim. Therefore, I strongly recommend the authors to clarify this point.

We have completely rewritten the paragraph explaining why we use 2 different predictors. Specifically, we argue that the difference between the average of one predictor and the average of another predictor corresponds roughly to the error between one predictor and experiments. However, by treating one predictor as giving rise to “experimental data” we can get access to both the average (as in experiments), but also the distribution of shifts (which we would not have been able to do if we had used real experimental data). If we instead had used a the same predictor for the target and the fitted ensembles, this would correspond to a predictor without any error. Such an analysis can give some insight but is not transferable to an experimental setup. We have extended the analysis by including a zero-error predictor to disentangle the contributions of the predictor error and ensemble differences. We have also re-written and extended the analysis of the uncertainties and their contribution to the reduced chi2 and to the fitting.

I could find some typos along the text:

lines 32-35:

“The ability to fold into different conformations allows IDPs to interact with different binding partners, and tuning their population these conformationsby post-translational modifications allows the regulation of their biological functions.”

line 284:

“...therefore, even after the reweighting, the helicity of this region cannot not increase and remains too...”

Captions of Figures S9 and S10 and in lines 475 and 477 of the manuscript: “...thinklines…”

We thank the reviewer for spotting these typos, which have been corrected.

Round 2

Reviewer 1 Report

The authors have satisfactorily addressed my concerns. I really appreciate the discussion about fitting and overfitting.

Reviewer 3 Report

Authors accomplished all the suggestions that I formulated in the previous report. Therefore, I recommend the publication of the article in its present form.